# Influence of Urbanization on Spatio-Temporal Characteristics of Extreme Hourly Precipitation in Shenyang

**Xue Ao [1], Qingfei Zhai [2], Chunyu Zhao [1],*, Yan Cui [1], Xiaoyu Zhou [1], Jingwei Li [1] and Mingqian Li [1]**

1    Shenyang Regional Climate Center of Liaoning, Shenyang 110166, China
2    Liaoning Provincial Meteorological Information Center, Shenyang 110166, China
*    Correspondence: springrainscn@163.com

**Abstract:** Understanding changes in extreme hourly precipitation is critical to urban planners for building more sustainable and resilient cities. In this study, we use satellite nighttime light data, urban land area data, population, and economic data to objectively classify urban and rural stations. Based on the hourly precipitation data from national meteorological stations in 1974–2020 and from regional automatic weather stations in 2005–2020 in Shenyang (China), the spatio-temporal distribution characteristics of the thresholds, maximums, intensities, and frequencies of extreme hourly precipitation (ExHP) in urban and rural areas are analyzed and compared. The results show that the large-value centers of ExHP thresholds, maximums, and intensities are mainly concentrated in urban areas. Both the frequency and intensity of ExHP at urban stations are obviously larger than those at rural stations, and the peaks mainly appear at night for stations of both two types. From 1974 to 2020, the average frequency and intensity of ExHP at urban stations both show increasing trends, with the increasing rate being much higher than those at rural stations. In terms of temporal variation, precipitation events of the abrupt type are the most frequent, accounting for 48.6% of the total, followed by the growing type (42.7%) and continuous type (8.7%). ExHP events of the abrupt type are mostly concentrated in Kangping County and Faku County of Northern Shenyang, but rarely occur in Xinmin City. ExHP events of the growing type are mainly found in Xinmin City and the municipal district of Shenyang. For urban stations, the ExHP frequency decreases in the early stage of urbanization, while increasing evidently during the rapid urbanization stage. However, the situation is just the opposite for rural stations. This indicates that the rapid urbanization in Shenyang has a certain impact on the increase in ExHP.

**Keywords:** urbanization; extreme hourly precipitation; Shenyang

## 1. Introduction

Under the background of global warming, the meteorological disasters in Liaoning Province (China) have been increasingly frequent, which brings a serious challenge to disaster prevention and mitigation [1,2]. From 2018 to 2019, 14 regional rainstorm processes occurred in Liaoning Province, with the emergency response to major meteorological disasters (rainstorms) being activated each time. A total of 14 cities were affected by the rainstorms, and the rainfall amount and rainfall days in many parts had set record highs, causing serious damages to transportation, animal husbandry, agriculture, and water conservancy projects. During 2018–2019, the number of typhoons affecting Liaoning Province is three times as many as that in the same period in history. These typhoons with strong intensity induced precipitation events with large rainfall amounts, which caused serious losses and continuously broke the extreme value of typhoon rainstorms [3]. On 16 August 2019 (local standard time, LST), the strongest short-term heavy rainfall in 68 years hit the municipal district of Shenyang with its intensity reaching the level of a heavy rainstorm, causing one death. From 1755 LST to 1855 LST, the precipitation observed at the meteorological station of Shenshuiwan Park in the Heping District reached

100.8 mm, which is the strongest hourly rainfall in the municipal district of Shenyang since complete meteorological observation records began in 1951. In August 2019, under the combined effect of Typhoon Lekima and the cold air brought by the upper-level trough, the regional-average precipitation amount in Liaoning Province reached 126.5 mm, breaking the record in the same period since 1951. This typhoon-induced precipitation event has been evaluated as a level 1 rainstorm disaster, which is the most serious for its long duration, large, accumulated rainfall amount, wide impact range, and accompanying strong gales. The rainstorm led to urban waterlogging in Shenyang, with the total area affected reaching 39.8 km$^2$ and a direct economic loss of 7.965 million yuan (1.669 million dollars) [4]. In addition, water conservancy projects and non-engineering measures for mountain torrents were also damaged to varying degrees. In recent decades, the impacts of rainstorm-induced mountain torrents, geological disasters, and urban waterlogging on Shenyang have become increasingly prominent. From 1984 to 2019, Shenyang suffered from varying degrees of meteorological disasters (including torrential rain, floods, and typhoons) year by year, resulting in a direct economic loss of more than 100 million yuan (14.65 million dollars) per year [5]. With the increasing population, social and economic development, and the continuous increases in carrying capacity and infrastructures per unit area in Shenyang, the damages caused by rainstorms are becoming more and more serious, resulting in great economic losses and a series of social and environmental problems. Hence, the rainstorm disaster has become a key factor restricting the sustainable development of Shenyang. Therefore, it is necessary to study the impact of urbanization on extreme heavy precipitation in this region.

According to the Clausius-Clapeyron equation, the water holding capacity of the atmosphere increases with the rising of atmospheric temperature. Therefore, the precipitation intensity will increase in the mid-latitudes and tropics which are both densely populated [6]. However, the precipitation intensity is usually expressed by daily precipitation data in previous studies, which may overestimate the intensity of long-term persistent weak precipitation and underestimate the intensity of short-term heavy precipitation, making the two types of precipitation events hard to be distinguished. Therefore, improving the temporal resolution of precipitation data is essential to studying extreme precipitation. In recent years, with the development of meteorological operation, a set of high-quality and relatively complete long-term hourly precipitation data has been established by Liaoning Meteorological Information Center, which is based on the observations at the national ground observation stations (61 stations in total, covering the national reference climatological stations, principal meteorological stations, and ordinary climatological stations) since records began and at the regional automatic stations (237 regional backbone stations in total) from 2005 to 2020 in Shenyang. This dataset provides important support for the study of extreme precipitation in Shenyang. So far, there have been some studies focusing on hourly precipitation in China. However, the research on extreme hourly precipitation (ExHP) events and urbanization effects is rarely seen. Hence, this dataset is essential to reveal the multi-scale spatio-temporal variation characteristics of ExHP in Shenyang.

Previous studies on extreme precipitation events and urbanization response are mainly concentrated in developed countries and highly developed regions in China, such as the Yangtze River Delta and Beijing-Tianjin-Hebei region. However, there are few studies on the urbanization effect of Shenyang, especially the impact of urbanization on extreme precipitation in this region. Studies have pointed out that the increase in urban buildings is conducive to convective precipitation and the enhancement of the urban heat island effect [7,8]. There is a significant relationship between the decrease in summer precipitation and the rapid urbanization in the Beijing-Tianjin-Hebei region [9–11]. Yang et al. [12,13] and Zhang et al. [14] suggested that the increasing trend of short-term heavy precipitation is closely related to urbanization. Moreover, the short-term heavy precipitation mostly occurs in the early morning, which may be caused by the joint action of the sea-land breeze and urbanization. Based on the TRMM (Tropical Rainfall Measuring Mission) satellite data, Li et al. [15] found that extreme heavy precipitation in the Pearl River Delta evidently

increases compared with that in other regions, while weak precipitation significantly reduces. Dai et al. [16] revealed that the summer precipitation intensity and the hours of extremely heavy precipitation in Liaoning Province both show an increasing trend, where the intensity of extreme precipitation in July greatly increases. According to multi-model simulation results, Miao et al. [17] found that the precipitation distribution in urban areas is closely related to the degree of urbanization. Liang and Ding [18] analyzed the 100-year hourly precipitation data in Shanghai and found that the hourly extreme precipitation in Shanghai has an obvious increasing trend. Especially, in the 30 years of rapid urbanization since the 1980s, the rain island effect is obvious in Shanghai, where the increase in the magnitude of extreme precipitation is more obvious in the city center than in the suburbs. In the research on the characteristics of summer precipitation in Beijing and the impact of urbanization on heavy precipitation, Yuan et al. [19] suggested that the contribution of extremely heavy precipitation in urban areas to the total summer precipitation has been increasing since 2004, which may be due to the urbanization effect. However, the previous works on urban precipitation are not specific enough. Most of the research is based on a few national meteorological stations, and the analysis on an hourly scale is also relatively few. Fewer studies may be caused by the lack of density and high-quality observational data in urban areas, which is a necessity for investigating the relations between urbanization and precipitation events at a finer spatial and temporal scale.

In this study, multi-source data including nighttime light data, socio-economic data, and urban area are used to determine the thresholds to distinguish between urban and rural areas, so as to classify the stations into urban and rural stations. The hourly precipitation data at national meteorological stations from 1974 to 2020 and regional automatic weather stations from 2005 to 2020 in Shenyang are also used. The percentile method is adopted to determine the threshold of ExHP at each station. On this basis, the spatio-temporal distribution characteristics of threshold, maximum, intensity, and frequency of ExHP at urban and rural stations in Shenyang are analyzed and compared, aiming to provide technical support for extreme precipitation prediction, meteorological disaster prevention and mitigation, and the adaptation to urban climate change in Shenyang.

The remainder of this paper is organized as follows. Section 2 introduces the data and method. Section 3 illustrates the spatial distribution characteristics of ExHP in Shenyang. Section 4 presents the time variation characteristics of ExHP, such as diurnal variation, interannual variation, and the variation characteristics of ExHP events of different types. Finally, the conclusions and discussion are provided in Section 5.

## 2. Data and Methods

### 2.1. Study Area

Shenyang is a high-density mega-city covering about 1.3 million square kilometers, which is located in the central part of Liaoning Province of Northeast China. It is dominated by plains and has a temperate monsoon climate with annual average temperature and precipitation of 8.4 °C and 690 mm, respectively. The rainfalls in Shenyang are mainly concentrated in summer. Shenyang has a larger temperature difference and obvious seasonal characteristics including hot-humid summer and cold-dry winter. Shenyang includes Kangping County, Faku County, Xinmin City, Liaozhong District, and the municipal district of Shenyang. Because of urban expansion, the population and economy in Shenyang grow rapidly, over half of the population lives in the municipal district of Shenyang. Subregions of Shenyang present different urbanization patterns with the varied densities of city population and buildings as well.

### 2.2. Research Data

This study uses five types of research data. First, the hourly temperature and precipitation data at 7 national meteorological stations during 1974–2020 and 237 regional automatic stations since records began until 2020 in Shenyang are adopted (Figure 1), which are provided by the Liaoning Meteorological Information Center. The national

meteorological stations are set up according to the national climate divisions to obtain long-term continuous climate data with sufficient representation. The number of stations is relatively small, while long-term observations. The regional automatic weather stations were established recently, but they provide more accurate and higher quality observations. They are thus an effective supplement to the national station data. Quality control procedures such as climatic extreme value and consistency tests have been applied to fix artificial errors. Second, the nighttime light data from the Defense Meteorological Satellite Program/Operational Line-Scan System (DMSP/OLS) with the study period during 1993–2013 are also utilized, which are provided by the National Geographic Data Center, https://www.ngdc.noaa.gov/eog/dmsp.html (accessed on 28 August 2022). The data is processed by accidental noise filtering and cloud elimination. Besides, it has also been specially developed with respect to the geographical characteristics of Asia, where the interference of light saturation is eliminated, and there is no necessity for manual control of the amplification gain. This data facilitates the analysis of urbanization intensity in Asia [20]. Third, the data on urban land area comes from the Liaoning Statistical Yearbook from 1993 to 2013. Fourth, the gross domestic product (GDP) per unit area adopts the 1-km grid GDP in China from the Resources and Environment Science and Data Center, http://www.resdc.cn (accessed on 28 August 2022). Fifth, the population density data comes from the Population Grid_China released by the Institute of Geographical Sciences and Natural Resources Research, Chinese Academy of Sciences. http://www.geodata.cn (accessed on 28 August 2022) . This dataset is established based on the land-use data from remote sensing data and population statistics. Through the spatial analysis function of the geographic information system, a spatial distribution model of the population is constructed to spatialize the statistical population data. Finally, the spatial population density data with a resolution of 1 km × 1 km is generated.

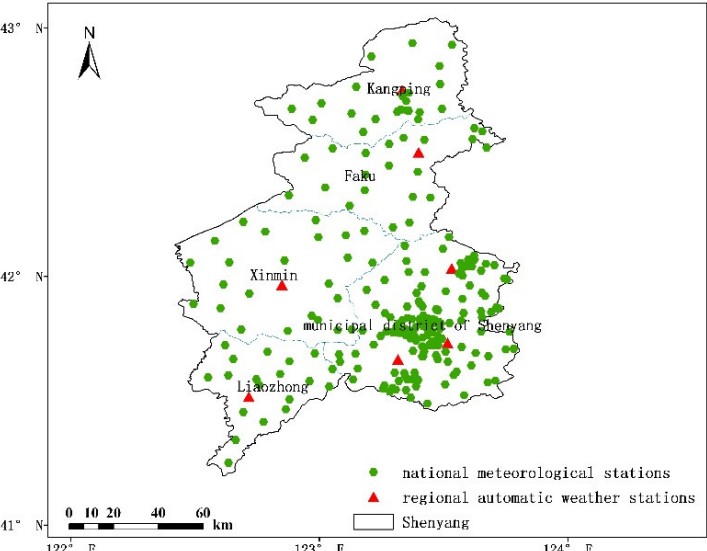

**Figure 1.** The spatial distribution of national meteorological stations and regional automatic weather stations in Shenyang.

## 2.3. Identification and Classification of Extreme Hourly Precipitation Events

In this study, the percentile method is used to determine the threshold of ExHP. The hourly precipitation data during 1974–2020 at each station are sorted from the weakest to the strongest. Through a comparison test, the hourly precipitation in the 99th percentile exceeding the minimum threshold of hourly heavy precipitation defined by the National Meteorological Center (20 mm·h$^{-1}$) is defined as the threshold of ExHP at this station [21]. The frequency of ExHP is defined as the occurrence number of hourly precipitation exceeding the above-defined threshold. The intensity of ExHP is defined as the ratio of the total amount of ExHP at each station to the frequency of the ExHP.

Based on the relationships of the hourly precipitation three hours before the occurrence of ExHP ($R_{-1}$, $R_{-2}$, $R_{-3}$) with the ExHP ($R_0$), the ExHP events from 1974 to 2020 are classified into the abrupt type, growing type, and continuous type [22]. ExHP events of the abrupt type generally last for a short time, with a small range and strong intensity. However, ExHP events of the continuous type are generally featured by longer duration and larger precipitation range with little variation in the intensity, mostly from the nimbostratus and altostratus and often related to the frontal system.

Specifically, when the magnitudes of $R_{-1}$, $R_{-2}$, and $R_{-3}$ are all less than one-tenth of that of $R_0$ (that is, $R_{-1} < 0.1 \times R_0$, $R_{-2} < 0.1 \times R_0$, and $R_{-3} < 0.1 \times R_0$), the precipitation event is defined as the abrupt type. When $R_{-1} > R_{-2}$ or $R_{-1} > R_{-3}$ or $R_{-2} > R_{-3}$ and at least one of ($R_{-1}$, $R_{-2}$, $R_{-3}$) has a magnitude greater than one-tenth of that of $R_0$ but smaller than $R_0$, it is defined as a growing type. When at least one of ($R_{-1}$, $R_{-2}$, $R_{-3}$) has a magnitude greater than that of $R_0$, it is defined as a continuous type [22].

The linear trend estimation is adopted for climate change analysis. The regression coefficient $b$ indicates the trend of climate variable $x$, where $b > 0$ indicates that $x$ exhibits an upward trend with the increase in time $t$ and $b < 0$ stands for the downward trend of $x$ with the increase in time $t$. The magnitude of $b$ reflects the changing rate, namely the degree of the upward or downward tendency [23].

In this study, the non-parametric statistical method is used for the significance test of linear trend [24], and the statistical formula is as follows:

$$Z = \frac{4}{n(n-1)} \sum_{i=1}^{n-1} r_i - 1 \tag{1}$$

$$Z_{0.05} = 1.96 \left[ \frac{4n + 10}{9n(n-1)} \right]^{\frac{1}{2}} \tag{2}$$

where $r_i$ in Equation (1) is the climatic sequence. If $|Z| > Z_{0.05}$, it is considered that the change trend is significant at the significance level of $\alpha = 0.05$.

## 2.4. Division of Urban and Rural Stations

By comparing with the statistical data, the spatial information of urban land use in Liaoning Province is extracted based on the DMSP/OLS nighttime light data [25–27]. The extraction mainly follows the two assumptions below. First, it is considered that the data on the urban land area in the statistical yearbook can accurately reflect the area of urban land in Shenyang, so the urban land area extracted from the DMSP/OLS data should be as close as possible to the data in the statistical yearbook. Second, the urban land area in Shenyang has expanded continuously since the 1990s. It is considered that the grid representing the urban area in the previous period does not disappear in the next period, that is, the grid in the previous period should be retained in the next period. On this basis, the dynamic thresholds are determined through the method of bisection, and the urban area under different thresholds is calculated. If the urban area calculated by the nighttime light data under a certain threshold is the closest to that in the statistical yearbook, this threshold is determined as the optimal threshold in Shenyang. After that, the average gray value of nighttime light within a radius of 7 km from each station is calculated. If the average value is greater than the threshold of the city where the station is located, this station is regarded as an urban station [28,29].

Figure 2 is a flowchart of the urban station and rural station divisions. D(i) is the threshold value of DMSP/OLS nighttime light data in a certain year (1 < D(i) < 63). Its initial value is set as 1 while i is the number of iterations. S(i) is the urban area of nighttime light of the city in this year when the threshold is equal to D(i), (S(i) is equal to the number of pixels × pixel area in an urban area). Area indicates the statistical value of the area of the city in the corresponding year, while P(i) is the average gray value of nighttime light within a radius of 7 km from each station.

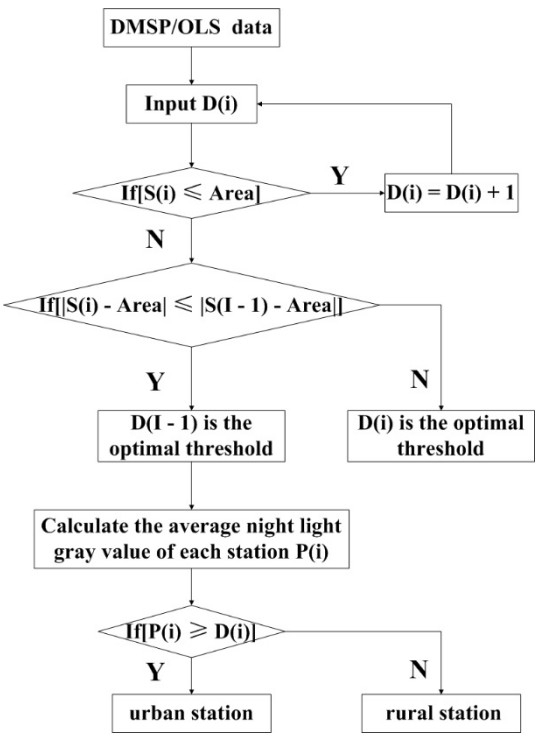

**Figure 2.** A flowchart of urban station and rural station division.

The spatial distribution of nighttime light gray values (Figure 3) shows that the gray values of some areas in the municipal district of Shenyang and Liaozhong District have increased from 1993 to 2003, indicating that these areas have undergone the urbanization process significantly. Combined with the population density data and the GDP per unit area (Figure 4), it can be seen that the urbanization process in Kangping County, Faku County, and Xinmin City is relatively slower. Therefore, the national meteorological stations and regional automatic weather stations in Liaozhong District and in the municipal district of Shenyang are selected as urban stations, and the rest of the stations are rural.

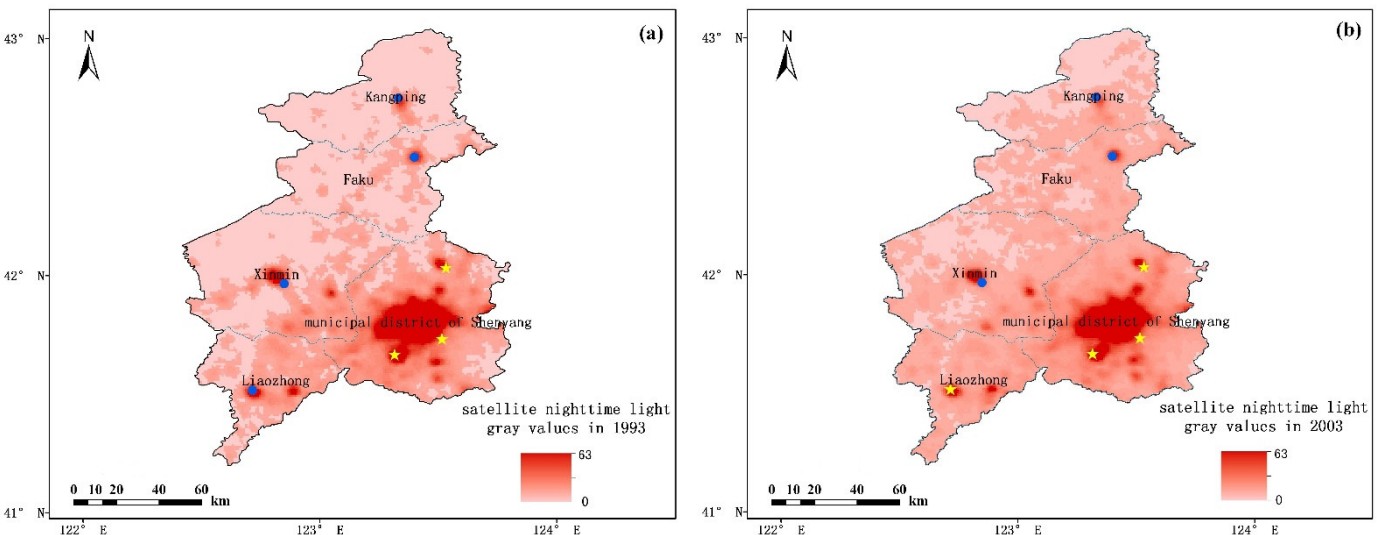

**Figure 3.** *Cont.*

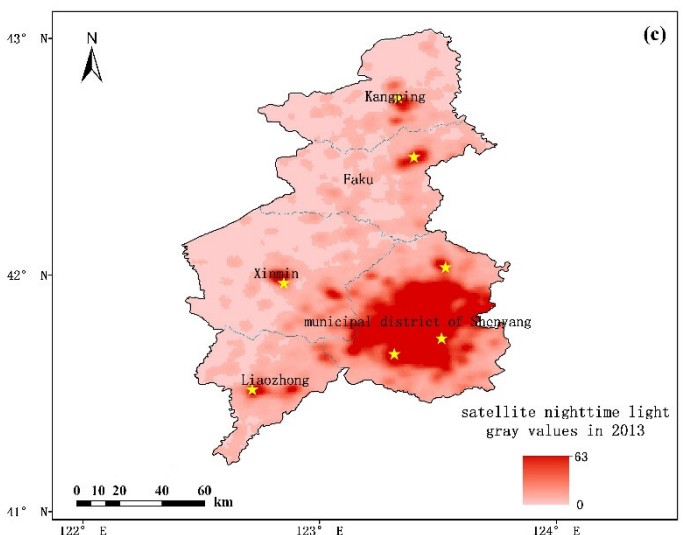

**Figure 3.** Spatial distributions of satellite nighttime light gray values in Shenyang in (**a**) 1993, (**b**) 2003, and (**c**) 2013 (yellow dots represent urban stations and blue dots represent rural stations).

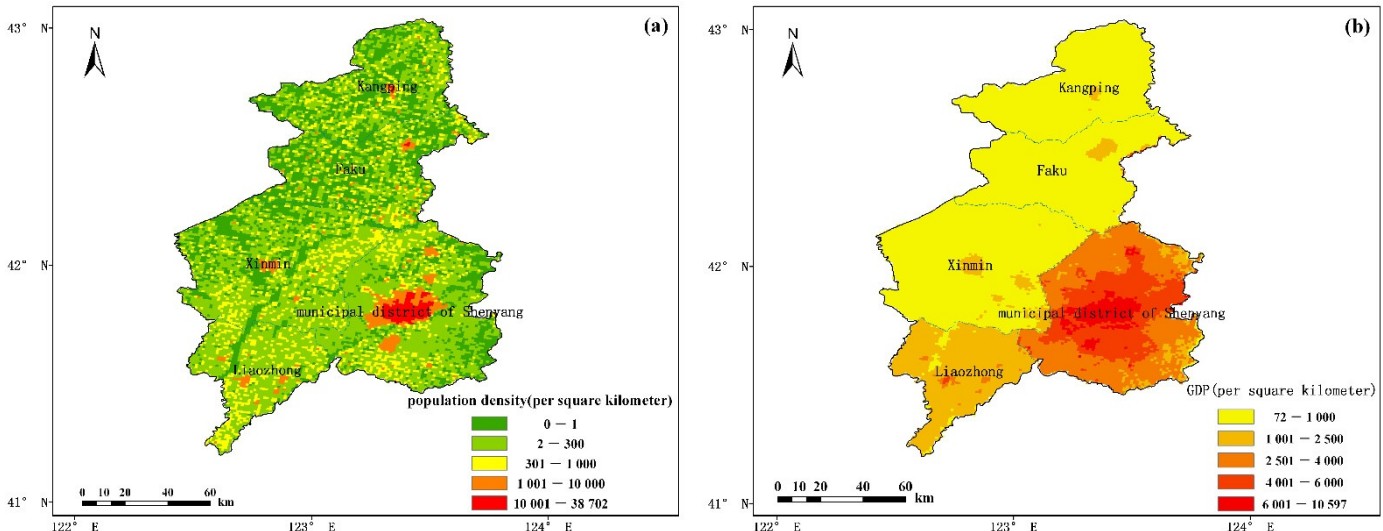

**Figure 4.** Spatial distributions of (**a**) population density and (**b**) gross domestic product per unit area in Shenyang.

### 2.5. Definition of Urbanization Impact

To quantitatively evaluate the urbanization impact on precipitation, the following terms are defined according to the Chu and Ren [30].

The linear trend of meteorological elements at stations near the city caused by urbanization factors is called the urbanization impact ($\Delta X_{ur}$), which is represented by the difference in the change trend of meteorological elements between urban stations ($X_u$) and rural stations ($X_r$):

$$\Delta X_{ur} = X_u - X_r \tag{3}$$

The contribution rate of urbanization impact refers to the proportion of urbanization impact to the change trend of meteorological elements at urban stations, which is expressed by $E_u$:

$$E_u = |\Delta X_{ur}/X_u| \times 100\% = |(X_u - X_r)/X_u| \times 100\% \tag{4}$$

## 3. Results

### 3.1. Spatial Distribution Characteristics of Extreme Hourly Precipitation in Shenyang

The spatial distributions of ExHP at national and regional stations (Figure 5) show that the large-value centers of the thresholds, maxima, and intensities of ExHP are mainly concentrated in the areas where urban stations are located. The maximum value of the ExHP threshold (32.43 mm·h$^{-1}$) is located at Beiling Park station in Huanggu District, followed by 29.9 mm·h$^{-1}$ at Zhujia Street station in Hunnan District. Table 1 reveals that all the large-value areas of ExHP thresholds are concentrated in the municipal district of Shenyang. Especially among the stations with the top 10 ExHP thresholds, 5 stations are located in the Sujiatun District. The areas with large ExHP thresholds are mainly located in both the southern parts of the municipal district of Shenyang and Liaozhong District. The maximum ExHP (115.7 mm·h$^{-1}$) is found in Kangping County at 1900 LST (Local Standard Time) on 16 July 1988, followed by 111 mm·h$^{-1}$ in Liaozhong District at 2100 LST on 3 July 1991, and 100.8 mm·h$^{-1}$ in Shenshuiwan Park of Heping District at 1800 LST on 16 August 2019. Combined with Table 1, it is revealed that the large values of ExHP maximums are mainly located in the municipal district of Shenyang, which mostly occur at night. Among the stations with the top 10 ExHP maximums, five stations are located in the municipal district of Shenyang and two in Liaozhong District. The large values of ExHP maximums are mainly concentrated in the southeastern part of the municipal district of Shenyang, western Liaozhong District, and northern Kangping County, while the values of ExHP maximums in Faku County and Xinmin City are relatively smaller. The maximum value of ExHP intensity (43.1 mm·h$^{-1}$) is located at Yangguang station in Yuhong District, followed by 42.3 mm·h$^{-1}$ at Beiling Park station of Huanggu District. Combined with Table 2, it can be seen that the large-value areas of ExHP intensity are all located in the municipal district of Shenyang. In addition, among the stations with the top 10 ExHP intensities, three stations are located in Huanggu District. Moreover, the spatial distribution also displays that the large values of ExHP intensity are mainly concentrated in the southeastern municipal district of Shenyang and the west part of Liaozhong District. In terms of spatial distribution, the ExHP frequency differs greatly from the threshold, intensity, and maximum. The ExHP frequency at urban stations is relatively low, whereas the large-value areas are mainly in Xinmin City. These may be due to the difference in construction time between national and regional stations. The hourly precipitation records at the national principal meteorological stations and reference climatological stations in Shenyang began in 1974, with few missing reports. However, the regional automatic weather stations have been gradually put into operation since 2003, which were established in different periods and have missing reports to varying degrees. Therefore, the top 10 ExHP frequencies are basically found at the national principal meteorological stations and reference climatological stations, and the maximum value (119 times) is located in Xinmin City.

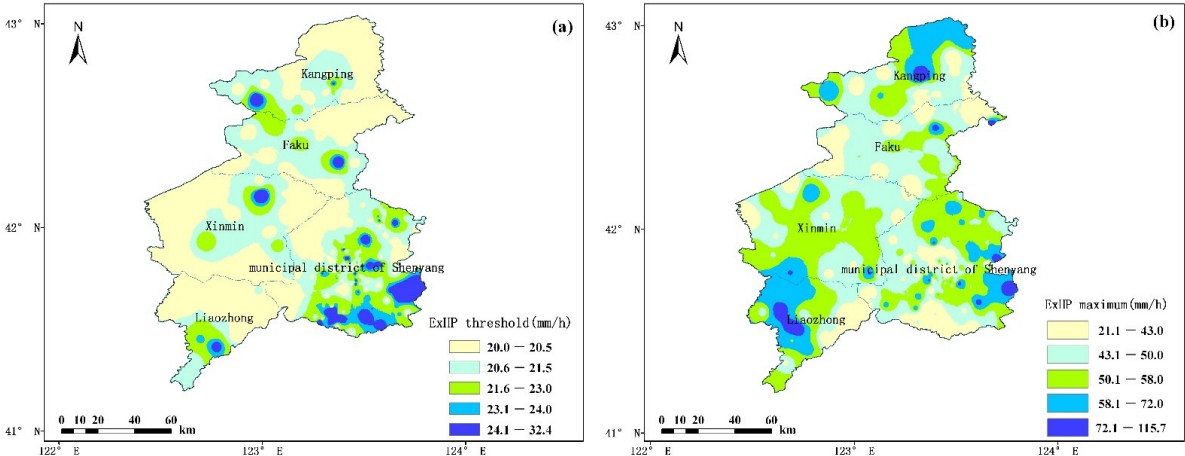

**Figure 5.** *Cont.*

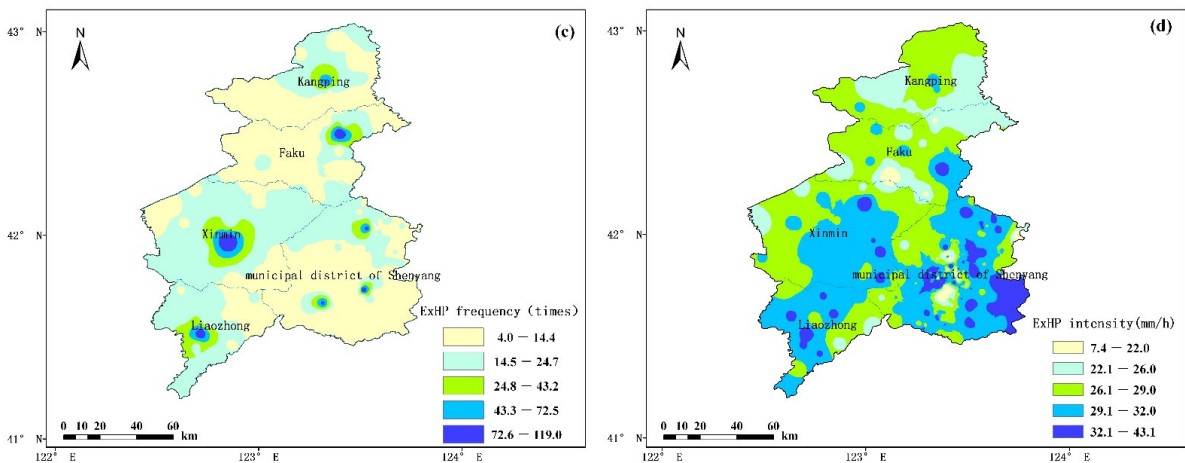

**Figure 5.** Spatial distributions of (**a**) extreme hourly precipitation (ExHP) threshold, (**b**) ExHP maximum, (**c**) ExHP frequency and (**d**) ExHP intensity.

**Table 1.** National and regional stations with the top 10 ExHP thresholds and maximums.

| County (District) | Station Name | Station Number | Threshold (mm·h⁻¹) | County (District) | Station | Station Number | Maximum (mm·h⁻¹) |
|---|---|---|---|---|---|---|---|
| Huanggu District | Beiling Park | L1138 | 32.43 | Kangping County | Kangping | 54244 | 115.7 |
| Hunnan District | Zhujia Street | L1089 | 29.901 | Liaozhong District | Liaozhong | 54332 | 111 |
| Sujiatun District | Linsheng Street | L1103 | 29.79 | Heping District | Shenshuiwan Park | L1310 | 100.8 |
| Sujiatun District | Linsheng Coal Mine | L1174 | 29.359 | Hunnan District | Gaokan Street | L1065 | 89.5 |
| Huanggu District | Santaizi Overpass | L1274 | 28.288 | Liaozhong District | Manduhu Town | L1100 | 86.7 |
| Hunnan District | Wangbin Street | L1085 | 28.04 | Faku County | Dangjiajie Village of Bojiagou Town | L1165 | 85 |
| Sujiatun District | Changxingdian Village of Linsheng Street | L1175 | 28 | Hunnan District | Wangbin Street | L1085 | 83.7 |
| Xinmin City | Gongzhutun Town | L1039 | 26.7 | Yuhong District | Yangguang | L1255 | 83.4 |
| Sujiatun District | Yantai Village of Bayihongling Street | L1170 | 26.43 | Hunnan District | Shenyang | 54342 | 82.2 |
| Sujiatun District | Chenxiang Street | L1109 | 26.375 | Faku County | Faku | 54245 | 79.8 |

**Table 2.** National and regional stations with the top 10 ExHP frequencies and intensities.

| County (District) | Station | Station Number | Frequency (Times) | County (District) | Station | Station Number | Intensity (mm·h⁻¹) |
|---|---|---|---|---|---|---|---|
| Xinmin City | Xinmin | 54333 | 119 | Yuhong District | Yangguang 100 | L1255 | 43.1 |
| Faku County | Faku | 54245 | 118 | Huanggu District | Beiling Park | L1138 | 42.3 |
| Hunnan District | Shenyang | 54342 | 117 | Tiexi District | Huaxiang Primary School | L1259 | 41.9 |
| Liaozhong District | Liaozhong District | 54332 | 106 | Shenhe District | Dongling Street | L1124 | 39.3 |
| Shenbei New District | Shenbei New District | 54248 | 91 | Hunnan District | Wangbin Street | L1085 | 38.3 |
| Sujiatun District | Sujiatun | 54340 | 82 | Shenfu Demonstration District | Shenjingzi Street | L1081 | 37.9 |
| Kangping County | Kangping | 54244 | 76 | Tiexi District | Nanba Bridge | L1126 | 37.7 |
| Liaozhong District | Yujiafang Town | L1120 | 24 | Huanggu District | Beitabeng | L1261 | 37.6 |
| Shenbei New District | Huishan Street | L1054 | 24 | Huanggu District | Santaizi Overpass | L1274 | 37.6 |
| Faku County | Yiniubaozi Town | L1034 | 24 | Shenhe District | Kepu Park | L1275 | 37.3 |

### *3.2. Temporal Variation Characteristics of Extreme Hourly Precipitation*

### 3.2.1. Diurnal Variation Characteristics

The diurnal variations of the regional-averaged ExHP frequency and intensity at urban and rural stations in Shenyang are shown in Figure 6. It is revealed that for urban stations, the diurnal variations of ExHP frequency and intensity both exhibit a bimodal distribution. The ExHP frequency peaks at 1800 LST and 2100 LST, while the ExHP intensity peaks in the afternoon (1300 LST) and at night (2100 LST). The ExHP frequency and intensity before dawn and in the early morning are significantly lower than those in the afternoon and at night. While for rural stations, the ExHP frequency and intensity mainly peak at night (2100 LST). It may be attributed to the enhancement of the southwest jet at night, which promotes water vapor transport, resulting in the frequent occurrence of short-term heavy rainfall at night [31]. To sum up, both the ExHP frequency and intensity at urban stations

are much larger than those at rural stations, and both urban and rural stations mainly peak at night.

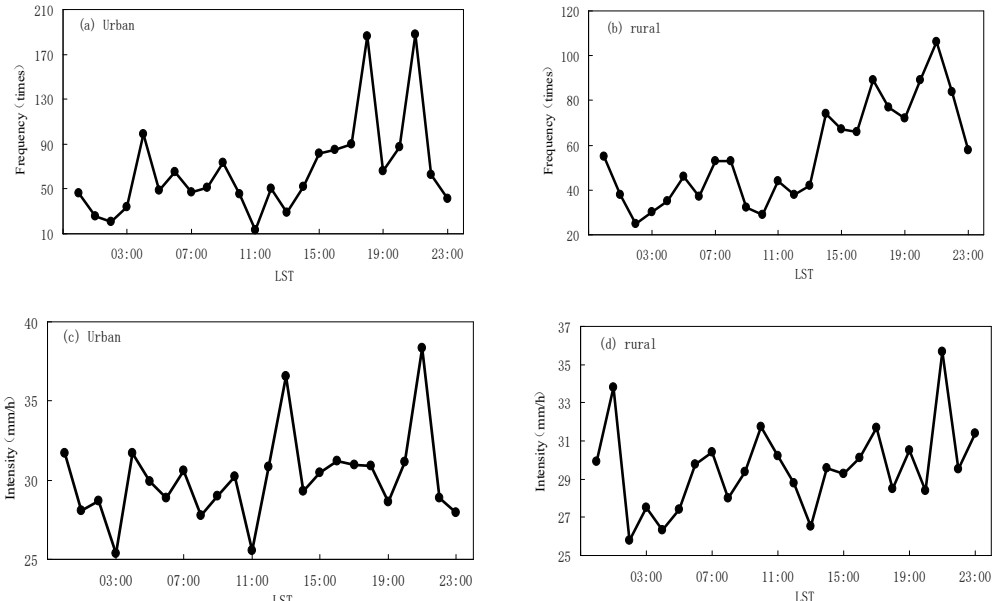

**Figure 6.** Diurnal variations of regional-averaged ExHP (**a**,**b**) frequency and (**c**,**d**) intensity at urban and rural stations in Shenyang.

### 3.2.2. Interannual Variation Characteristics

Figure 7 displays that both the ExHP frequency and intensity averaged at urban stations in Shenyang show increasing trends from 1974 to 2020, which are 0.17 times·10a$^{-1}$ and 0.76 mm·h$^{-1}$·10a$^{-1}$, respectively, where the variation trend of ExHP intensity has passed the significance test. However, there is no obvious variation trend of regional-averaged ExHP frequency at rural stations, and the ExHP intensity shows an upward trend with the trend value being 0.41 mm·h$^{-1}$·10a$^{-1}$. It can be seen that the increasing rate of ExHP intensity at urban stations is about twice that of rural stations. For both ExHP frequency and intensity, the increasing rates at urban stations are obviously larger than those at rural stations. The changes at rural stations are relatively gentle, indicating that rural stations can well represent the change of background field and the developing rate of areas not affected by urbanization, and thus the selection of rural stations is reasonable. Table 3 shows the urbanization effect on precipitation in Liaoning Province from 1974 to 2020. It can be found that urbanization has a great impact on precipitation intensity, and the contribution rate of urbanization reaches 46.1%, indicating that the increase in ExHP intensity in Liaoning Province in the last 47 years might be attributed to the impact of rapid urbanization.

**Table 3.** Urbanization impact and its contribution rate (%) on annual extreme hourly precipitation frequency and intensity at urban and rural stations in Liaoning Province from 1974 to 2020.

| Type of Station | ExHP Frequency (Times·(10a)$^{-1}$) | ExHP Intensity (mm·h$^{-1}$·(10a)$^{-1}$) |
|---|---|---|
| Urban station | 0.17 | 0.76 * |
| Rural station | −0.004 | 0.41 * |
| Urbanization impact | 0.174 | 0.35 * |
| Contribution rate of Urbanization impact (%) | - | 46.1 |

Note: "*" means that the value has passed the significance test at 0.05 significance level; "-" means that the urbanization impact fails the significance test at 0.05 significance level, and then the contribution rate of urbanization impact will not be calculated.

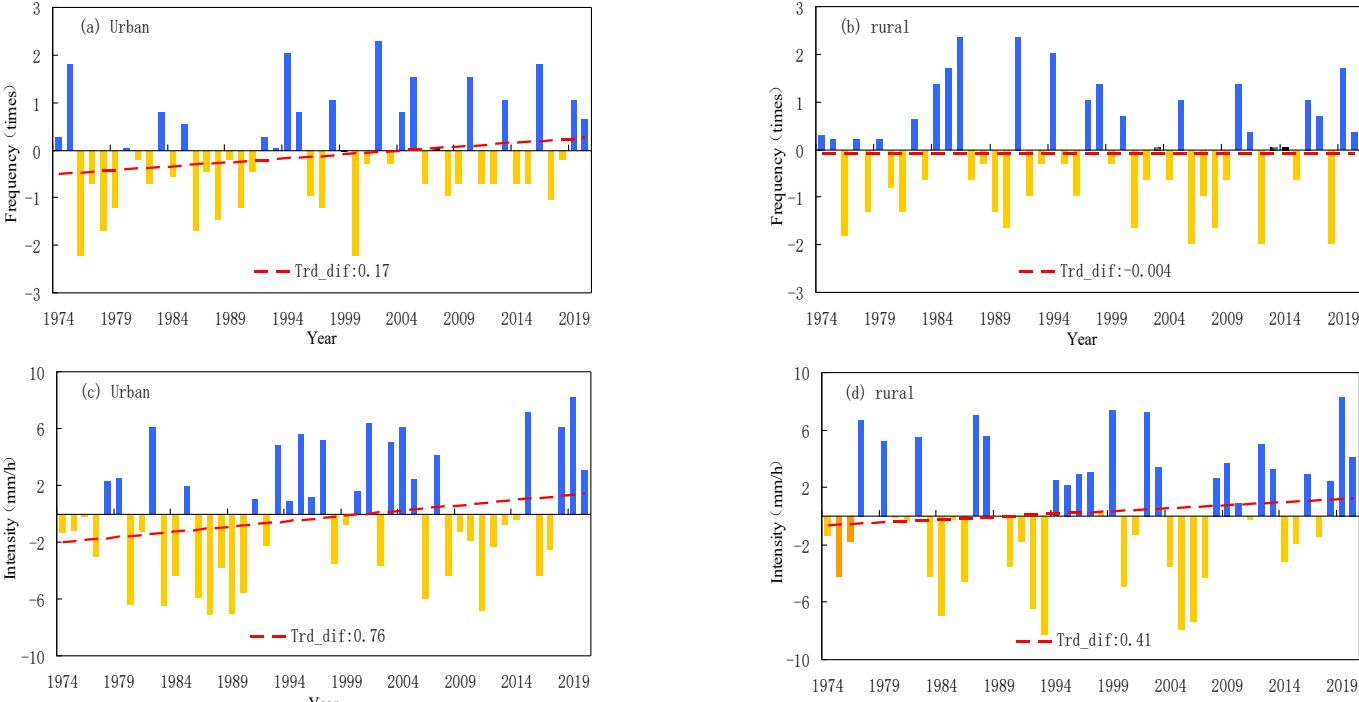

**Figure 7.** Interannual variations of regional-averaged ExHP (**a**,**b**) frequency and (**c**,**d**) intensity at urban and rural stations in Shenyang from 1974 to 2020.

To explore the variation of ExHP during the urbanization process in Shenyang, it is necessary to divide the development process in Shenyang into different periods. Economic development and population growth in urban regions are the main signs of urbanization. The statistical yearbooks show that the growth rate of the non-agricultural population in Shenyang is 2.7%·$(10a)^{-1}$ from 1974 to 1990. After the 1990s, the non-agricultural population increases continuously with the rapid development of the urban social economy in Shenyang. From 1991 to 2020, the growth rate of the non-agricultural population in Shenyang reaches 9.78%·$(10a)^{-1}$, and meanwhile, the GDP begins to grow rapidly from the 1990s. Therefore, the period before the early 1990s is selected as the early stage of urbanization in Shenyang in this study, while the period after the 1990s is the rapid development stage. The ExHP frequency at urban stations basically decreases in the early stage of urbanization (1974–1990), while increases evidently in the rapid development stage of urbanization (1991–2020) (Figure 8a–d). The ExHP frequency at rural stations increases in the early stage but decreases in the rapid development stage (Figure 8e–g). From 1974 to 2020, the change rates at the four urban representative stations are 0.47 times·$(10a)^{-1}$, 0.25 times·$(10a)^{-1}$, 0.55 times·$(10a)^{-1}$ and 0.41 times·$(10a)^{-1}$, respectively. Meanwhile, the change rates at the three rural representative stations are −0.02 times·$(10a)^{-1}$, 0.14 times·$(10a)^{-1}$, and −0.03 times·$(10a)^{-1}$, respectively. Meanwhile, a sensitivity experiment is carried out on the selected time nodes (1991 mentioned above) through a comparison between the previous and subsequent periods. It is found that the conclusions are basically the same no matter which year from 1990 to 1993 is selected as the time node. This indicates that the conclusions can truly reflect the differences in the variation characteristics of ExHP in different urbanization stages. In a word, the rapid urbanization in Shenyang has a certain impact on the increase in ExHP.

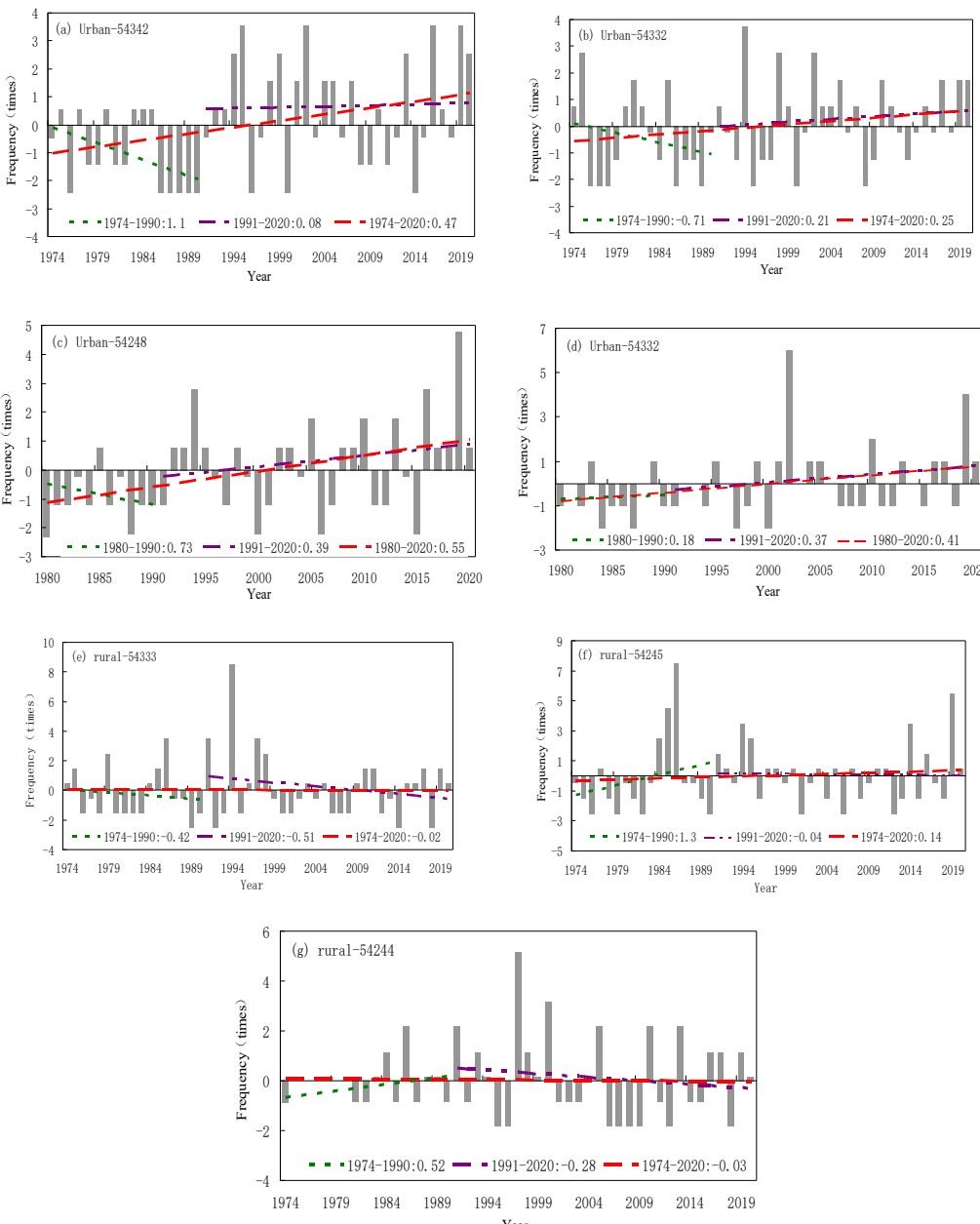

**Figure 8.** Annual ExHP frequencies at urban representative stations (**a–d**) and rural representative stations (**e–g**) in Shenyang from 1974 to 2020.

### 3.2.3. Variation Characteristics of Different Types of Extreme Hourly Precipitation Events

According to the evolution characteristics of precipitation three hours before the occurrence of ExHP, ExHP events are divided into the abrupt type, growing type, and continuous type (Table 4). Based on the frequencies of different types of ExHP events averaged in the whole of Shenyang City from 1974 to 2020, ExHP events of the abrupt type are the most, accounting for 48.6% of the total events, followed by the growing type (42.7%). There are relatively few ExHP events of the continuous type, accounting for only 8.7%.

Figure 9 shows the spatial distributions of the proportions of three types of ExHP events at national and regional stations. It is revealed that the abrupt type and growing type present opposite spatial distribution patterns. ExHP events of the abrupt type are prone to occur in Kangping and Faku Counties of northern Shenyang, but rarely in Xinmin City. However, ExHP events of the growing type occur more frequently in Xinmin City and the municipal district of Shenyang, but less in Kangping and Faku Counties. While ExHP

events of the continuous type are mainly concentrated in northeastern Xinmin City and Hunnan District, but rarely occur in Kangping County, Faku County, Tiexi District, and Sujiatun District.

**Table 4.** Classification criteria and proportions of different types of ExHP events in Shenyang.

| Type | Classification Criteria | Proportion |
|---|---|---|
| abrupt type | $R_{-1} < 0.1 \times R_0$, $R_{-2} < 0.1 \times R_0$ and $R_{-3} < 0.1 \times R_0$ | 48.6% |
| growing type | When $R_{-1} > R_{-2}$ or $R_{-1} > R_{-3}$ or $R_{-2} > R_{-3}$ and at least one of ($R_{-1}$, $R_{-2}$, $R_{-3}$) has a magnitude greater than one-tenth of that of $R_0$ but smaller than $R_0$ | 42.7% |
| continuous type | When at least one of ($R_{-1}$, $R_{-2}$, $R_{-3}$) has a magnitude greater than that of $R_0$ | 8.7% |

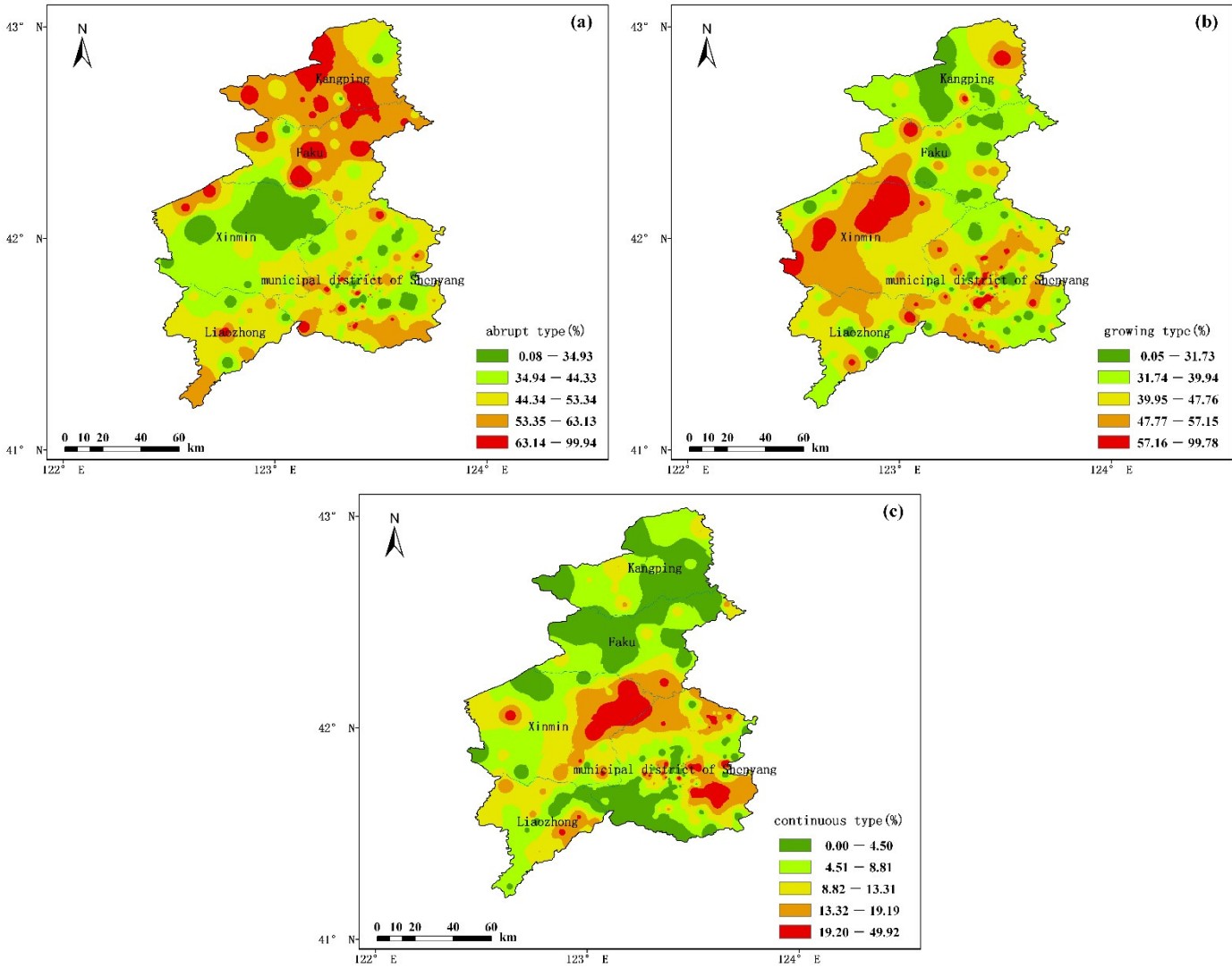

**Figure 9.** Spatial distributions for ExHP events of the (**a**) abrupt type, (**b**) growing type, and (**c**) continuous type in Shenyang.

## 4. Discussion

In this study, we evaluated the ExHP changes using the multi-source datasets by classifying the urban and rural stations, and in turn investigating the influences of urbanization on the ExHP. Results clearly indicate that the ExHP for urban and rural stations peaks in the nighttime, while urban stations have larger ExHP frequency and intensity than rural stations. It may be attributed to the enhancement of the southwest jet at night, which can promote water vapor transport and result in the frequent occurrence of short-term heavy

rainfall at night [31]. Relatively, due to the diurnal variation of solar radiation heating, the low-level atmosphere is easy to be unstable during the afternoon and evening, benefiting local moist convection activities and short-term heavy precipitation. Meanwhile, the strong nighttime heat island effect produced by rapid urbanization, combined with the slowdown of the movement of the weather system by buildings and the increase of condensation nuclei, have enhanced nighttime precipitation [32–34]. Some studies believe that the nighttime precipitation peak comes from stratiform cloud precipitation, which is strengthened by the instability caused by the nighttime radiative cooling of the cloud top, the accumulation of water vapor at the low level greatly promotes the development of nighttime convection [35,36].

This study found that the large-value centers of ExHP thresholds, maximums, and intensities are mainly located in urban areas. Both the ExHP frequency and intensity averaged at urban stations in Shenyang show increasing trends from 1974 to 2020, which may be related to the urbanization of Shenyang. Li et al. [37] found that there are obvious high-value centers in the urban area and its vicinities, such as the hourly precipitation, precipitation intensity, and the frequency of short-term heavy precipitation in Beijing in summer, which is consistent with the results in this paper. Liu [38] also pointed out the main urban area of Beijing has an obvious "urban rain island effect" from July to August and the hourly precipitation and precipitation intensity are significantly greater than those in the suburbs. Yin et al. [39] pointed out that the increase in summer precipitation in the urban area of Beijing may be caused by the combined effect of the urban heat island circulation and the wind in the valley. Local sensible and latent heat transport in urban areas enhances geopotential instability in the lower atmosphere, directly triggering convection [40]. Surface temperature plays the most important role in enhancing sensible heat transport and triggering precipitation, and the heterogeneity of urban underlying surfaces enhances the locality of precipitation [41]. The water vapor of local convective precipitation comes from the evaporation of the surface and the upward transport of water vapor when the low-level wind field converges [42]. The impact of cities on climate is not limited to the above aspects. The urbanization effect changes the local circulation in the city and in turn affects the distribution and intensity of precipitation [43].

Since this study is focused on urbanization rates affect ExHP, it might be interesting to study how this is likely to evolve in the next decades. Some studies have used spatial and transient modelling of land use/land cover (LULC) to simulate and predict the changes in land use/land cover and population growth in the future. The model can predict spatial and temporal land changes for the next decades. It can also be a framework to understand underlying drivers of changes [44]. In the next step, we can use the model to assess the urbanization rates of Shenyang in the future, which can be used to predict the trend of ExHP and provide recommendations for urban planning.

A recent study using WAF model simulations shows that differences in the timing for urban-induced rainfall in the inland versus coastal environment, and the shape of a city/urban area can significantly affect and increase rainfall in this city. A circular city shows nearly 22.0% larger rainfall accumulation and 78.6% greater rainfall intensity compared to a triangular city over urban surfaces. The rainfall anomaly is caused by different urban-rural circulations over various city shapes. The strong low-level convergence over circular cities favors efficient upward moisture transport and deep convection [45]. The municipal district of Shenyang is a circular inland urban area, this study explains the reason for the increase in ExHP in the urban area. In future research work, we need to further explore the processes and mechanisms of the interaction between urbanization and precipitation through climate models and other technical methods. In order to provide technical support for extreme precipitation prediction, meteorological disaster prevention and mitigation, and the adaptation to urban climate change.

## 5. Conclusions

The large-value centers of ExHP thresholds, maximums, and intensities are mainly located in urban areas. The maximum ExHP threshold appears at Beiling Park station in Huanggu District with the value being 32.43 mm·h$^{-1}$. The maximum ExHP is 115.7 mm in Kangping County. The maximum ExHP intensity is 43.1 mm·h$^{-1}$ at Yangguang station in Yuhong District. The top 10 ExHP frequencies mainly occur at national principal meteorological stations and reference climatological stations, and the maximum value (119 times) appears in Xinmin City. The diurnal variations of ExHP frequency and intensity at urban stations both exhibit a bimodal distribution. The ExHP frequency and intensity before dawn and in the early morning are significantly lower than those in the afternoon and at night. Both the ExHP frequency and intensity at urban stations are much larger than those at rural stations, and both urban and rural stations mainly peak at night. From 1974 to 2020, both the ExHP frequency and intensity averaged at urban stations in Shenyang show increasing trends, with the increasing rate being obviously larger at urban stations than at rural stations.

According to the statistical yearbooks, the period before the early 1990s is selected as the early stage of urbanization in Shenyang, and the period after the 1990s is the rapid development stage. Thus, the ExHP frequency at urban stations basically decreases in the early stage of urbanization and increases evidently in the rapid development stage of urbanization. While the ExHP frequency at rural stations increases in the early stage of urbanization but decreases in the rapid development stage. The contribution rate of urbanization on ExHP intensity reaches 46.1%, indicating that rapid urbanization has a certain impact on the increase in ExHP in Liaoning Province.

From 1974 to 2020, ExHP events of the abrupt type are the most, accounting for 48.6% of the total ExHP events, followed by the growing type (42.7%). While there are relatively few ExHP events of the continuous type, accounting for only 8.7%. ExHP events of the abrupt type are prone to occur in Kangping and Faku Counties of northern Shenyang, but rarely in Xinmin City. However, ExHP events of the growing type occur more frequently in Xinmin City and the municipal district of Shenyang. Besides, ExHP events of the continuous type are mainly concentrated in northeastern Xinmin City and Hunnan District.

**Author Contributions:** Conceptualization, X.A. and C.Z.; methodology, Q.Z.; formal analysis, Y.C.; data curation, X.Z.; writing—original draft preparation, X.A.; writing—review and editing, Q.Z.; visualization, J.L.; supervision, M.L. All authors have read and agreed to the published version of the manuscript.

**Funding:** This research was funded by the Scientific research project of liaoning Meteorological Bureau in 2021 (No. 202108) and the Science and Technology Plan Project in Liaoning Province (No. 2019-MS-199).

**Institutional Review Board Statement:** Not applicable.

**Informed Consent Statement:** Not applicable.

**Data Availability Statement:** Some models, or code that support the findings of this study are available from the corresponding author upon reasonable request.

**Conflicts of Interest:** The authors declare no conflict of interest.

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
