# Peer review of "Influence of Urbanization on Spatio-Temporal Characteristics of Extreme Hourly Precipitation in Shenyang"

_land, doi:10.3390/land11091487_

Round 1

Reviewer 1 Report

The study "Influence of Urbanization on Spatio-Temporal Characteristics of Extreme Hourly Precipitation in Shenyang" analyzed the spatiotemporal distribution characteristics of the thresholds, maximums, intensities, and frequencies of extreme hourly precipitation (ExHP) based on the hourly precipitation data from national meteorological stations in 1974–2020 and automatic regional weather.

Overall, the research theme is reasonable and within the journal's scope. However, in this present form, this work required solid literature support (comment 2). and significant revisions for the discussion section (Comment 6). Therefore, this work can be considered for a detailed report after gentle modifications.

1.      The abstract of this work should be reconstructed. Consider reporting a few words about the topic background at the start instead of directly reporting the objectives. Also, consider reporting the main highlights of the adopted methodology.

2.      The number of references used in this work is too low; From Line 28-82, the authors reported only one citation. All information/statements can be considered valid/authentic with a single citation. The authors discussed all the statements there without any reference/.citation. Then just 14 references for 2.5 pages introduction. Without citation, it seems that all authors' perceptions are based on which they generate their arguments. Authors should completely reconstruct the introduction section scientifically.

***Currently, I am not making any comment on that inferior introduction section. However, it would be further considered after a complete and gentle modification.

3.      The quality of the figure (1, 2, 3, 4, and 8) used in this work is too low and hard to read and understand (with unclear legends). I do not know whether the quality is inferior from the author's side or it has been decreased because of conversion.

4.      Consider providing "Study Area details." and study area map.

5.      Section 2.1. Consider reporting the references/sources for the utilized datasets.

6.      Provide a detailed methodology flowchart to understand the flow of methods.

7.      Although the findings of this research are intriguing, inferior Discussion makes it duller and doesn't provide enough support to the author's conclusions. Therefore, the discussion section should be revised entirely by arguing based on previous studies and the author's findings.

Author Response

Dear expert, 

Thanks very much for taking your time to review this manuscript. I really appreciate all your comments and suggestions! We have tried our best to revise the manuscript according to your kind and construction comments and suggestions. We sincerely hope that this revised manuscript has addressed all your comments and suggestions.

Here follows the detailed response to you. Please see the attachment.

Kind regards,

Ms. Ao

Reviewer 2 Report

The authors provided a very interesting study and analysis of the influence of Urbanization on Spatio-Temporal Characteristics of Extreme Hourly Precipitation in Shenyang in China. The study design is sound, and the findings are important and well presented. I believe the study has potential for publication in the Land Journal.

A major and general comment is that all the Figures resolution should be improved. Also, I have adressed several other comments based on expert knowledge, and a wealth of suggestions to improve the manuscript, attached to this message.

I suggest Minor Revision.

Author Response

(The authors gave the same response as above.)

Round 2

Reviewer 1 Report

The authors have addressed all my suggestions, and the manuscript is now significantly improved and acceptable for publication.